# Functional Autonomy Evaluation Levels in Middle-Aged and Older Spanish Women: On Behalf of the Healthy-Age Network

**Pablo Jorge Marcos-Pardo** [1,2,*], **Noelia González-Gálvez** [1,2,*], **Raquel Vaquero-Cristóbal** [1,2],
**Gemma María Gea-García** [1,2], **Abraham López-Vivancos** [1,2], **Alejandro Espeso-García** [1,2],
**Daniel Velázquez-Díaz** [2,3,4], **Ana Carbonell-Baeza** [2,3,4], **David Jiménez-Pavón** [2,3,4],
**Juliana Brandão Pinto de Castro** [5] **and Rodrigo Gomes de Souza Vale** [2,5,6]

1   Research Group on Health, Physical Activity, Fitness and Motor Behaviour (GISAFFCOM) and Physical
    Activity and Sport Sciences Department, Faculty of Sport, Catholic University San Antonio of Murcia,
    30107 Murcia, Spain; rvaquero@ucam.edu (R.V.-C.); gmgea@ucam.edu (G.M.G.-G.);
    alvivancos@ucam.edu (A.L.-V.); aespeso@ucam.edu (A.E.-G.)
2   Active Aging, Exercise and Health/HEALTHY-AGE Network, Consejo Superior de Deportes (CSD),
    Ministry of Culture and Sport of Spain, 28040 Madrid, Spain; daniel.velazquez@uca.es (D.V.-D.);
    ana.carbonell@uca.es (A.C.-B.); david.jimenez@uca.es (D.J.-P.); rodrigovale@globo.com (R.G.d.S.V.)
3   MOVE-IT Research Group, Department of Physical Education, Faculty of Education Sciences,
    University of Cádiz, 11519 Cádiz, Spain
4   Biomedical Research and Innovation Institute of Cádiz (INiBICA) Research Unit, Puerta del Mar University
    Hospital, University of Cádiz, 11519 Cádiz, Spain
5   Laboratory of Exercise and Sport (LABEES), Rio de Janeiro State University (UERJ), 20550 Rio de Janeiro,
    Brazil; julianabrandaoflp@hotmail.com
6   Exercise Physiology Laboratory, Estacio de Sá University, 22640-102 Cabo Frio, Brazil
*   Correspondence: pmarcos@ucam.edu (P.J.M.-P.); ngonzalez@ucam.edu (N.G.-G.)

**Abstract:** Aging is associated with a progressive loss of functional capacity that affects the health and quality of life of middle-aged and older people. The purpose of this study was to report functional autonomy evaluation levels in middle-aged and older women in the Spanish context. A total of 709 middle-aged and older women, between 50 and 90 years old, were selected to participate in the study. The sample was divided by age category every five years. The functional autonomy levels were determined by the Latin American Group for Maturity (GDLAM) protocol and we developed a classification pattern for middle-aged and older women living in Spain. The GDLAM Index (GI) was then calculated to assess functional autonomy. The classification of the tests and the GI followed the percentile rank (P) Very Good ($p < 0.15$), Good ($p$ 0.16–$p$ 0.50), Regular ($p$ 0.51–$p$ 0.85), and Poor ($p > 0.85$). It was considered that the lower the value found for the percentile, the better the result. The GDLAM protocol showed strong reliability with intraclass correlation coefficient (ICC) values greater than 0.92 in all tests. It is observed that all variables of the GDLAM protocol presented a positive and significant correlation with age ($p < 0.001$). The Roc Curve showed that GI values higher than 26 (CI95% = 0.97–1.00; $p < 0.001$) and 32 (CI95% = 0.98–1.00; $p < 0.001$) for middle-aged and elderly women, respectively, can predict and indicate low functional autonomy. The normative values hereby provided will enable evaluation and adequate interpretation of Spanish middle-aged and older women's functional autonomy.

**Keywords:** functional capacity; older people; functional autonomy assessment; aging; health

## 1. Introduction

Aging is associated with a decrease in the efficiency of several physiological processes, including a progressive loss of functional capacity. The functional decrease tends to occur earlier in women, mainly around menopause [1], involving postural control instabilities, which lead to changes in walking and balance [2]. Besides balance, other components of physical fitness that influences health are muscle strength, aerobic capacity, and flexibility in adults as well in older people [3,4]. These physical capabilities are directly related to the performance of activities of daily living (ADL) and prevention and control of non-communicable diseases [3,5]. Thus, functional autonomy is one of the most relevant markers related to health, quality of life, and performance of ADL of adults and older people [6].

The Latin American Group for Maturity (GDLAM) defines autonomy from three perspectives: autonomy of action, which refers to physical independence; autonomy of will, which means the possibility of self-determination; autonomy of thoughts, which enables the individual to judge any situation. According to GDLAM, independence is the ability to perform tasks without help of people, devices, or systems [7]. The decline in functional autonomy is one of the main consequences of aging and can lead to frailty of older individuals [8,9]. Frailty occurs in a multifactorial manner as a result of different physiological regressive processes associated with aging [10]. This progressive deterioration, together with the decrease in strength and endurance, places the individual in a vulnerable condition of functional capacity and quality of life [11]. Quality of life is one of the factors responsible for the population's longevity. Maintaining lifestyle habits that promote functional autonomy in older individuals requires an awareness effort. Hence, physical activities are important to achieve the desired standard in certain features of quality of life and functional autonomy in these individuals [12]. Specifically in Spain, population projections estimate that, in the coming decades, the population aged 65 and over will continue to increase. The projection of older people for 2060 is one-third of the total Spanish population, which corresponds to over 16 million older people, being women are in the majority, outnumbering men by 32% [13]. Furthermore, in recent years, the physical inactivity levels of Spanish young populations have increased [14], which brings a red alert as age-related declines in functional capacity are clearly accentuated by physical inactivity [15], thus these inactivity levels in the young increase the risk of dependence later in life [16].

Therefore, more research on ADL performance of older people, especially in women as an understudied population, can give health professionals precise standards to classify functional autonomy and verify exercise training program efficiency [17]. One instrument used in many studies [18–25] to evaluate the functional autonomy levels in older people is the GDLAM protocol. This protocol measures the time it takes, in seconds, to complete five typical ADL [17]. Then, the current study aims to determine the functional autonomy levels using the GDLAM protocol and to develop a classification pattern for middle-aged and older women living in Spain.

## 2. Materials and Methods

### 2.1. Participants

The study initially counted 1054 older adults from Social Care Programs for older people in Murcia and Cádiz, Spain, from April to November 2019. These individuals passed through the following inclusion criteria: being female; being 50 years of age or older; being independent in their ADL; could be doing physical exercise or not. Any type of acute or chronic condition that could compromise or become an impediment factor for the performance of functional autonomy tests was considered an exclusion criterion, such as: cardiopathies, diabetes, hypertension, and uncontrolled bronchitis as well as any musculoskeletal conditions that could be an interaction factor for the tests (osteoarthritis, recent fracture, tendinitis, and prosthesis use); neurological problems; morbid obesity; chronic renal individuals and those who used medication that could cause attention disorders.

After the end of the sampling process, 709 older women were selected to participate in the study. The participants, who ranged between 50 and 90 years old, were divided by age category every five years.

The survey participants signed the informed consent term according to the Helsinki Declaration [26]. The study was approved by the Ethics and Research Committee involving Human Beings of the Catholic University San Antonio of Murcia under protocol n. CE031907.

*2.2. Data Collection Procedures*

2.2.1. Anthropometric Evaluation

A mechanical scale with a capacity of 150 kg and a precision of 100 g, with a stadiometer, from Filizola (Brazil), was used for the evaluation of body mass and height, following the protocol of the International Society for the Advancement of Kinanthropometry [27] for body mass index (BMI) calculation.

2.2.2. Functional Autonomy Evaluation

Functional autonomy was assessed through the Latin American Group for Maturity (GDLAM) protocol of autonomy [7,17,25,28] composed of the following tests:

(1) Walk 10 m (W10 m): the purpose of this test is to evaluate the gait speed of the individual for a distance of 10 m [29] (Figure 1a).

(2) To sit and get up from the chair and move around the house (SCMA): the objective is to assess the ability of the middle-aged and older people in their agility and balance in life situations. With a chair fixed on the ground, two cones should be demarked diagonally to the chair, at a distance of four meters behind and three meters to the right and left sides of the same. The individual begins the test seated in the chair, with her feet on the floor, and with the sign "already," she gets up, goes to the right, moves around the cone, returns to the chair, sits down, and takes both feet off the ground. Without hesitating, she does the same move to the left. Immediately, the same course is again completed, to the right and left, thus making the entire journey and circulating each cone twice, in the shortest time possible [30] (Figure 1b).

(3) Stand up from sitting position (SSP): the test aims to assess the functional capacity of the lower limbs. The test starts with the individual in the sitting position in a chair without arm support, and the seat at a distance from the ground of 50 cm, then, the individual stands up and takes a seat five times consecutively [31] (Figure 1c).

(4) Standing up from the prone position (SPP): the purpose of this test is to assess the overall ability of the individual to get up the floor. The test starts with the individual in a ventral decubitus position, with arms along the body; the command of "now", indicates that the individual must get up, leaving the position as soon as possible [32] (Figure 1d).

(5) To put on and take off a T-shirt (PTS): the individual should be standing with arms along the body and a T-shirt in one of the hands. At the voice signal of "Go," the individual should put on the shirt and immediately take it off, returning to the starting position. This test is intended to measure the agility and coordination of the upper limb [33] (Figure 1e).

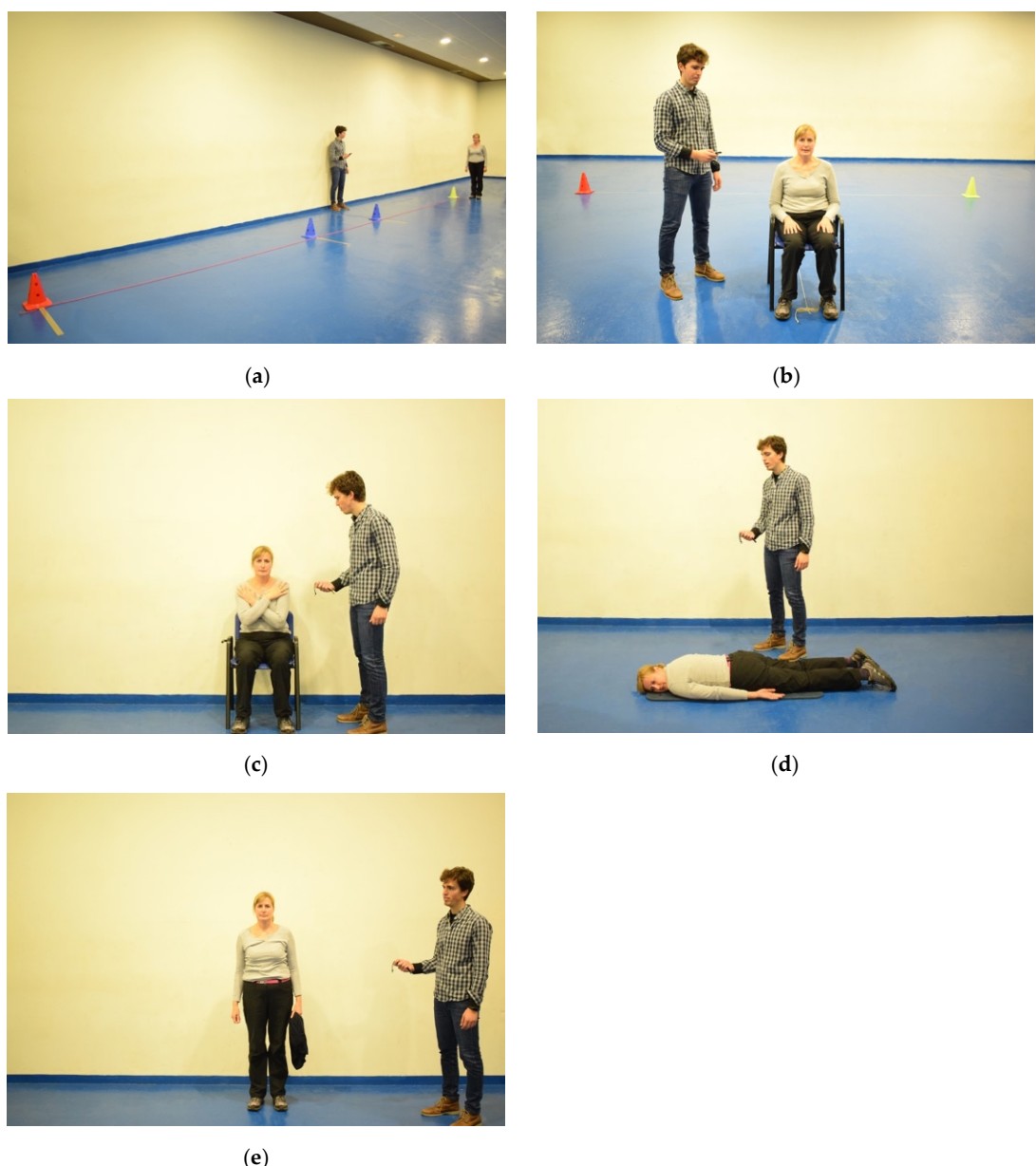

**Figure 1.** Latin American Group for Maturity (GDLAM) protocol tests in starting position. (**a**) Walk 10 m (W10 m); (**b**) To sit and get up from the chair and move around the house (SCMA); (**c**) Stand up from sitting position (SSP); (**d**) Standing up from the prone position (SPP); (**e**)To put on and take off a T-shirt (PTS).

All tests were performed in two attempts for each individual in a suitable environment, with a minimum interval of five minutes, in which the shortest time in seconds was recorded through a stopwatch. The equipment used consisted of a 48 cm chair (measured from the seat to the floor), a stopwatch (Casio, Malaysia), two cones, a T-shirt, a mat (Olive Fitness, Spain), and a sunny brand metal tape measure. After these tests, the GDLAM index of autonomy (GI) was calculated in scores [16,28], where the lower the value of the score, the better the result, using the following formula. All the tests were measured using time in seconds.

$$GI = [(W10 \text{ m} + SSP + SPP + PTS) \times 2] + SCMA]/4$$

Validity and Reliability

The functional autonomy GDLAM protocol underwent a validity analysis using the face validity method [7,17]. A panel composed of three doctors who are specialists in aging and two doctors who are specialists in measures and evaluations, who were not related to the present study to avoid any influence on their opinions, unanimously approved the protocol, with 100% agreement, as a valid instrument to assess the functional autonomy related to the performance of the older people's ADL, applicable to the population of Spain. Before the initial data collection, two experienced evaluators applied the GDLAM protocol tests to 30 older women randomly selected on three different occasions with a minimum interval of 72 h between them to calculate the intraclass correlation coefficient (ICC). The results showed strong reliability with ICC values greater than 0.92 in all tests. One-way ANOVA was used to compare the measurements obtained between the tested occasions and did not show significant differences ($p < 0.05$).

*2.3. Statistical Analysis*

The data were expressed as mean, standard deviation, and percentage values. The coefficient of variation was applied to analyze the dispersion of the sample data. The normality of the data was verified through the Box-Cox method. The classification levels of the tests and the GI were established by age group every five years and from 50 years of age on, using the percentile (*p*). It was considered that the lower the value found for the percentile the better the result, according to the classification: Very Good ($p < 0.15$), Good ($p$ 0.16–$p$ 0.50), Regular ($p$ 0.51–$p$ 0.85), and Poor ($p > 0.85$). The one-way analysis of variance (ANOVA) was used, followed by Tukey's post hoc, to identify possible differences in tests and in GI between age groups. Pearson's correlation test was applied for the analysis of associations between age and GDLAM protocol variables. The cut-off threshold for satisfactory GI was determined by the Youden's J statistic (Youden's index) [34,35], this index corresponding values of sensitivity and specificity for groups G1, G2, and G3 (middle-aged women) and G4, G5, G6, and G7 (older women) of the studied population and calculated from the analysis of the receiver operator characteristics (ROC) curve. The study admitted the value of $p < 0.05$ for statistical significance. The data were processed by IBM SPSS Statistics 23 (IBM SPSS, Inc., Chicago, IL, USA).

**3. Results**

Table 1 presents the anthropometric characteristics of the sample by age categories every 5 years. It is observed that the highest concentration of participants is found in G4 and G5.

**Table 1.** Characteristics of the sample by age groups.

| Age Groups | N | % | Age (Years) | Body Mass (kg) | Height (m) | BMI |
|---|---|---|---|---|---|---|
| G1 (50–54 years) | 37 | 5.22 | 51.59 ± 1.26 | 77.18 ± 16.13 | 1.65 ± 0.10 | 28.21 ± 4.08 |
| G2 (55–59 years) | 36 | 5.08 | 56.67 ± 1.35 | 77.57 ± 12.50 | 1.65 ± 0.10 | 28.36 ± 3.40 |
| G3 (60–64 years) | 70 | 9.87 | 61.69 ± 1.54 | 76.77 ± 14.64 | 1.63 ± 0.09 | 28.72 ± 4.41 |
| G4 (65–69 years) | 283 | 39.92 | 66.40 ± 1.39 | 69.93 ± 11.55 | 1.56 ± 0.08 | 28.58 ± 4.20 |
| G5 (70–74 years) | 160 | 22.57 | 71.91 ± 1.45 | 73.08 ± 11.71 | 1.57 ± 0.09 | 29.52 ± 4.08 |
| G6 (75–79 years) | 86 | 12.13 | 76.09 ± 1.22 | 73.62 ± 13.29 | 1.59 ± 0.10 | 29.13 ± 4.26 |
| G7 (≥80 years) | 37 | 5.22 | 84.13 ± 3.20 | 72.57 ± 9.10 | 1.54 ± 0.08 | 30.71 ± 4.55 |
| Total | 709 | 100 | | | | |

BMI: body mass index.

Table 2 presents the performance of the tests and the GI by age categories.

**Table 2.** Sample results by age groups.

| Groups | Tests | Average | Classification |
|---|---|---|---|
| G1 (n = 37; 50–54 years) | W10 m | 5.35 ± 1.45 | Regular |
| | SSP | 9.38 ± 1.71 | Regular |
| | SPP | 2.99 ± 1.22 | Regular |
| | PTS | 11.64 ± 2.83 | Good |
| | SCMA | 34.72 ± 6.07 | Regular |
| | GI | 23.36 ± 3.80 | Regular |
| G2 (n = 36; 55–59 years) | W10 m | 5.85 ± 1.73 | Regular |
| | SSP | 9.48 ± 2.66 | Regular |
| | SPP | 3.02 ± 1.25 | Regular |
| | PTS | 12.00 ± 2.28 | Regular |
| | SCMA | 36.11 ± 7.04 | Regular |
| | GI | 24.20 ± 4.72 | Regular |
| G3 (n = 70; 60–64 years) | W10 m | 6.72 ± 1.69 | Regular |
| | SSP | 11.24 ± 3.94 | Regular |
| | SPP | 4.05 ± 1.91 | Regular |
| | PTS | 13.45 ± 3.64 | Regular |
| | SCMA | 40.98 ± 7.93 | Regular |
| | GI | 27.97 ± 5.82 | Regular |
| G4 (n = 283; 65–69 years) | W10 m | 6.95 ± 1.52 | Regular |
| | SSP | 11.98 ± 3.19 | Regular |
| | SPP | 5.62 ± 3.22 | Good |
| | PTS | 15.00 ± 6.12 | Regular |
| | SCMA | 42.01 ± 5.78 | Regular |
| | GI | 30.28 ± 5.09 | Regular |
| G5 (n = 160; 70–74 years) | W10 m | 7.32 ± 1.73 | Regular |
| | SSP | 12.17 ± 3.39 | Regular |
| | SPP | 6.46 ± 3.56 | Regular |
| | PTS | 15.17 ± 5.03 | Regular |
| | SCMA | 45.37 ± 9.57 | Regular |
| | GI | 31.90 ± 6.42 | Regular |
| G6 (n = 86; 75–79 years) | W10 m | 7.88 ± 1.69 | Regular |
| | SSP | 12.68 ± 4.63 | Regular |
| | SPP | 6.83 ± 3.43 | Regular |
| | PTS | 17.44 ± 6.47 | Regular |
| | SCMA | 48.63 ± 8.48 | Regular |
| | GI | 34.57 ± 6.29 | Regular |
| G7 (n = 37; ≥80 years) | W10 m | 8.49 ± 2.70 | Regular |
| | SSP | 13.50 ± 3.61 | Regular |
| | SPP | 6.90 ± 3.41 | Regular |
| | PTS | 20.52 ± 9.08 | Regular |
| | SCMA | 56.28 ± 14.49 | Regular |
| | GI | 38.77 ± 8.66 | Regular |

W10 m: walk 10 m; SSP: stand up from sitting position; SPP: standing up from the prone position; PTS: to put on and take off a T-shirt; SCMA: to sit and get up from the chair and move around the house; GI: GDLAM autonomy index.

Some fluctuations of the coefficient of variation (CV%) are observed across the age categories for the different tests, with small increases in SPP and PTS with the age, especially in G7 category (Figure 2).

Figure 2 shows the evolution of the coefficient of variation (CV%) across age categories by GDLAM tests.

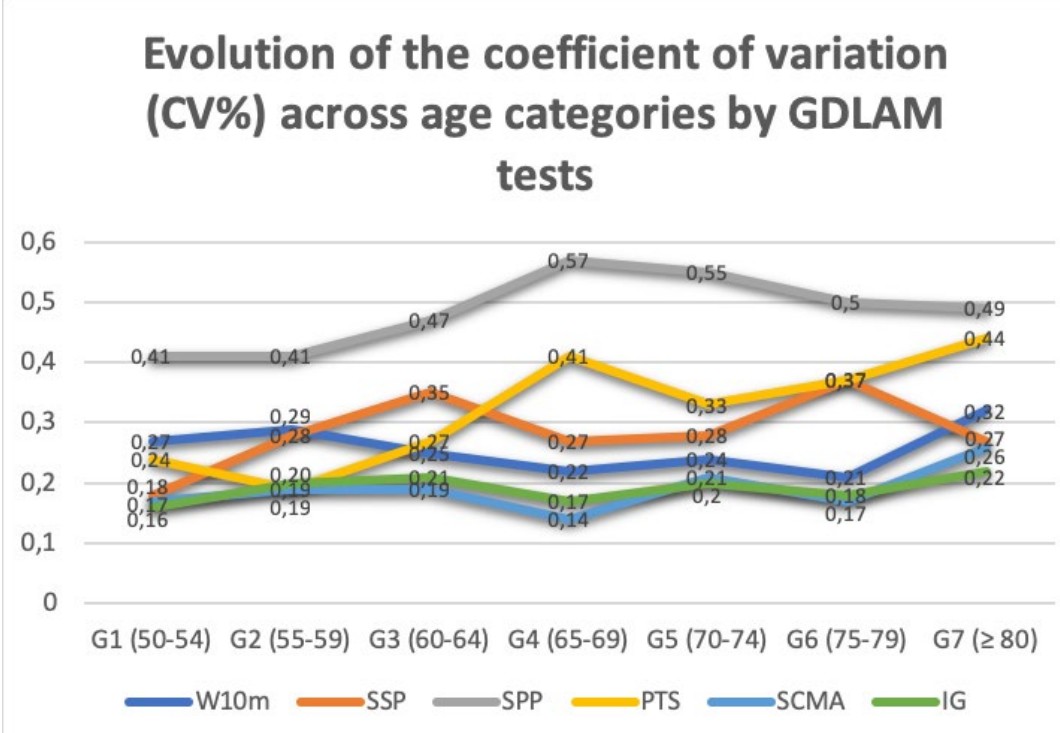

**Figure 2.** Evolution of the coefficient of variation (CV%) across age categories and test.

Figure 3 shows the distribution of GI scores by age categories and percentile. The GI values raise as the age category increase. It is observed that the tendency curves are similar among all age categories.

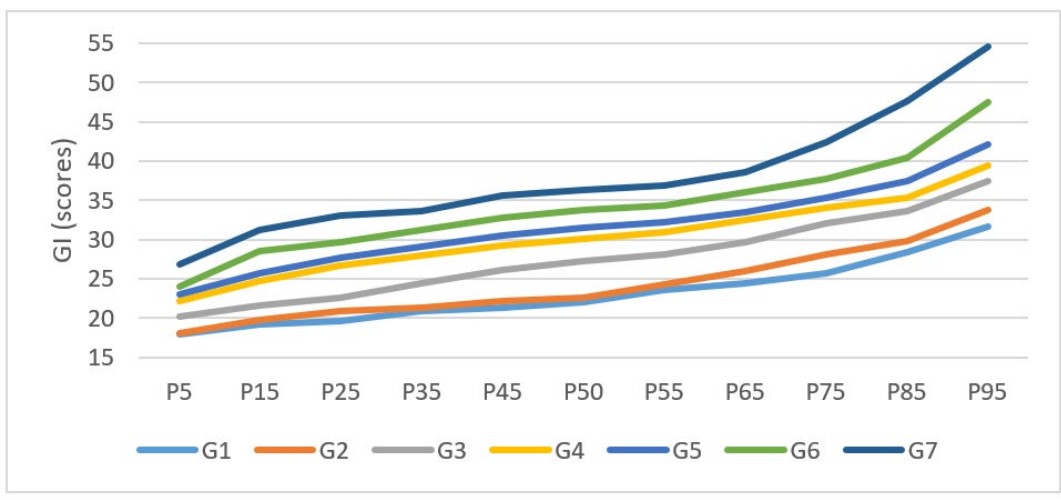

**Figure 3.** GI Percentile Distribution by Age Categories.

Table 3 presents the comparative results of the tests and the GI of the GDLAM protocol between the age groups. It was observed that the G7 showed longer execution times in the tests W10 m, SSP, SPP, PTS, SCMA, and GI when compared to the groups G1, G2, G3, G4, and G5 ($p < 0.05$). G7 presented lower performance ($p < 0.05$) in PTS, SCMA, and lower GI compared to G6. No significant differences were found between G7 and G6 in the tests W10 m, SSP, and SPP and between G1 and G2 in all tests and in the GI.

**Table 3.** Comparative analysis of the GDLAM protocol between age groups.

| | G1 (50–54 yrs) | G2 (55–59 yrs) | G3 (60–64 yrs) | G4 (65–69 yrs) | G5 (70–74 yrs) | G6 (75–79 yrs) | G7 (≥80 yrs) |
|---|---|---|---|---|---|---|---|
| W10 m | 5.35 ± 1.45 *§†‡ | 5.85 ± 1.73 *§†‡ | 6.72 ± 1.69 *§ | 6.95 ± 1.52 *§ | 7.32 ± 1.73 * | 7.88 ± 1.69 | 8.49 ± 2.70 |
| SSP | 9.38 ± 1.71 *§†‡∂ | 9.48 ± 2.66 *§†‡∂ | 11.24 ± 3.94 *§† | 11.98 ± 3.19 * | 12.17 ± 3.39 * | 12.68 ± 4.63 | 13.50 ± 3.61 |
| SPP | 2.99 ± 1.22 *§†‡ | 3.02 ± 1.25 *§†‡ | 4.05 ± 1.91 *§†‡ | 5.62 ± 3.22 *§ | 6.46 ± 3.56 | 6.83 ± 3.43 | 6.90 ± 3.41 |
| PTS | 11.64 ± 2.83 *§†‡∂ | 12.00 ± 2.28 *§†‡ | 13.45 ± 3.64 *§†‡ | 15.00 ± 6.12 *§ | 15.17 ± 5.03 *§ | 17.44 ± 6.47 * | 20.52 ± 9.08 |
| SCMA | 34.72 ± 6.07 *§†‡∂ | 36.11 ± 7.04 *§†‡∂ | 40.98 ± 7.93 *§† | 42.01 ± 5.78 *§† | 45.37 ± 9.57 *§ | 48.63 ± 8.48 * | 56.28 ± 14.49 |
| GI | 23.36 ± 3.80 *§†‡∂ | 24.20 ± 4.72 *§†‡∂ | 27.97 ± 5.82 *§†‡ | 30.28 ± 5.09 *§† | 31.90 ± 6.42 *§ | 34.57 ± 6.29 * | 38.77 ± 8.66 |

W10 m: walk 10 m; SSP: stand up from sitting position; SPP: standing up from the prone position; PTS: to put on and take off a T-shirt; SCMA: to sit and get up from the chair and move around the house; GI: GDLAM autonomy index. * $p < 0.05$ for G7; § $p < 0.05$ for G6; † $p < 0.05$ for G5; ‡ $p < 0.05$ for G4; ∂ $p < 0.05$ for G3.

Table 4 shows the correlation coefficient between age and functional autonomy, both of the tests and GI. It is observed that all variables of the GDLAM protocol of autonomy presented a positive and significant correlation with age. This shows that the higher the score, the lower the functional autonomy in relation to the tests (in relation to time) and the GI (in relation to the scores). Therefore, in the present study, the higher the age, the longer the time to complete the autonomy tests and the higher the GI score.

**Table 4.** Correlation analysis between age and functional autonomy (tests and GI).

|  |  | W10 m | SSP | SPP | PTS | SCMA | GI |
|---|---|---|---|---|---|---|---|
| Years | r | 0.362 | 0.227 | 0.227 | 0.301 | 0.481 | 0.466 |
|  | *p*-value | <0.001 | <0.001 | <0.001 | <0.001 | <0.001 | <0.001 |

W10 m: walk 10 m; SSP: stand up from sitting position; SPP: standing up from the prone position; PTS: to put on and take off a T-shirt; SCMA: to sit and get up from the chair and move around the house; GI: GDLAM autonomy index.

Table 5 presents a classification pattern of the GDLAM protocol of functional autonomy for each age group. The classification was made by separate tests and by GI for each age category. It is observed that the lower the values of the tests and the GI, the better the classification.

**Table 5.** Standardized GDLAM protocol classification of functional autonomy for residents in Spain.

| Age Groups |  | Very Good | Good | Regular | Poor |
|---|---|---|---|---|---|
| G1 (n = 37; 50–54 years) | W10 m | <4.17 | 4.17–4.85 | 4.86–7.18 | >7.18 |
|  | SSP | <7.06 | 7.06–9.01 | 9.02–11.13 | >11.13 |
|  | SPP | <1.77 | 1.77–2.61 | 2.62–4.85 | >4.85 |
|  | PTS | <9.95 | 9.95–11.73 | 11.74–14.65 | >14.65 |
|  | SCMA | <29.19 | 29.19–32.87 | 32.88–41.41 | >41.41 |
|  | GI | <19.16 | 19.16–22.11 | 22.12–28.35 | >28.35 |
| G2 (n = 36; 55–59 years) | W10 m | <4.24 | 4.24–5.52 | 5.53–8.00 | >8.00 |
|  | SSP | <7.60 | 7.60–9.30 | 9.31–12.19 | >12.19 |
|  | SPP | <1.80 | 1.80–2.87 | 2.88–4.24 | >4.24 |
|  | PTS | <8.44 | 8.44–10.83 | 10.84–15.45 | >15.45 |
|  | SCMA | <30.80 | 30.80–35.26 | 35.27–45.67 | >45.67 |
|  | GI | <19.77 | 19.77–22.68 | 22.69–29.81 | >29.81 |
| G3 (n = 70; 60–64 years) | W10 m | <5.00 | 5.00–6.65 | 6.66–8.49 | >8.49 |
|  | SSP | <8.13 | 8.13–9.87 | 9.88–16.36 | >16.36 |
|  | SPP | <2.16 | 2.16–3.36 | 3.37–6.21 | >6.21 |
|  | PTS | <9.18 | 9.18–13.19 | 13.20–17.76 | >17.76 |
|  | SCMA | <32.45 | 32.45–39.90 | 39.91–49.80 | >49.80 |
|  | GI | <21.66 | 21.66–27.23 | 27.24–33.62 | >33.62 |
| G4 (n = 283; 65–69 years) | W10 m | <5.40 | 5.40–6.80 | 6.81–8.73 | >8.73 |
|  | SSP | <8.89 | 8.89–11.61 | 11.62–15.50 | >15.50 |
|  | SPP | <3.03 | 3.03–6.26 | 6.27–9.58 | >9.58 |
|  | PTS | <9.97 | 9.97–14.48 | 14.49–20.19 | >20.19 |
|  | SCMA | <36.03 | 36.03–41.50 | 41.51–48.03 | >48.03 |
|  | GI | <24.79 | 24.79–30.11 | 30.12–35.40 | >35.40 |
| G5 (n = 160; 70–74 years) | W10 m | <5.50 | 5.50–7.12 | 7.13–8.91 | >8.91 |
|  | SSP | <8.92 | 8.92–11.91 | 11.92–15.04 | >15.04 |
|  | SPP | <3.63 | 3.63–4.91 | 4.92–7.90 | >7.90 |
|  | PTS | <10.49 | 10.49–13.79 | 13.80–20.08 | >20.08 |
|  | SCMA | <37.22 | 37.22–44.43 | 44.44–51.35 | >51.35 |
|  | GI | <25.73 | 25.73–31.49 | 31.50–37.50 | >37.50 |
| G6 (n = 86; 75–79 years) | W10 m | <6.14 | 6.14–7.45 | 7.46–9.45 | >9.45 |
|  | SSP | <9.15 | 9.15–12.13 | 12.14–16.90 | >16.90 |
|  | SPP | <3.85 | 3.85–5.36 | 5.37–9.60 | >9.60 |
|  | PTS | <11.87 | 11.87–15.30 | 15.31–24.22 | >24.22 |
|  | SCMA | <41.52 | 41–52–47.18 | 47.19–57.89 | >57.89 |
|  | GI | <28.50 | 28.50–33.83 | 33.84–40.46 | >40.46 |
| G7 (n = 37; ≥80 years) | W10 m | <6.12 | 6.12–7.94 | 7.95–11.16 | >11.16 |
|  | SSP | <9.87 | 9.87–13.48 | 13.49–16.33 | >16.33 |
|  | SPP | <4.41 | 4.41–5.90 | 5.91–10.67 | >10.67 |
|  | PTS | <12.43 | 12.43–18.47 | 18.48–29.53 | >29.53 |
|  | SCMA | <42.46 | 42.46–52.67 | 52.68–75.04 | >75.04 |
|  | GI | <31.28 | 31.28–36.30 | 36.31–47.65 | >47.65 |

W10 m: walk 10 m; SSP: stand up from sitting position; SPP: standing up from the prone position; PTS: to put on and take off a T-shirt; SCMA: to sit and get up from the chair and move around the house; GI: GDLAM autonomy index. Tests: values are in seconds; GI: values are units of the scores.

Figure 4A,B shows the analysis of the ROC curve for the prediction of middle-aged (G1, G2, and G3: 50–64 years) and older individuals (G4, G5, G6, and G7: ≥65 years) with low functional autonomy. The GI cutoff points >28 and >34 showed strong sensitivity (99.8) and specificity (89.6) for the group of middle age and strong sensitivity (99.7) and specificity (87.6) for the group of older women.

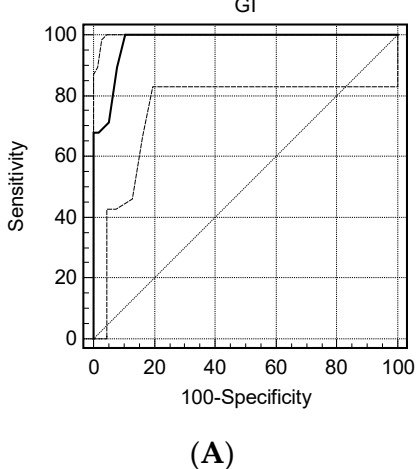

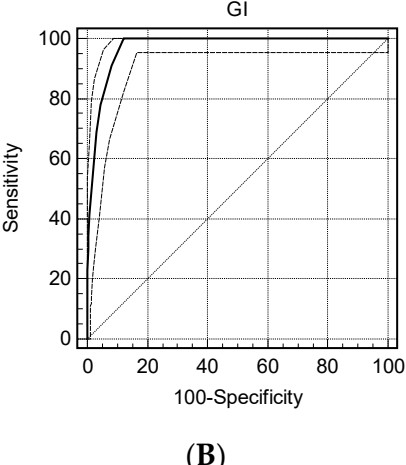

**(A)**    **(B)**

**Figure 4.** (**A**,**B**). Analysis of the ROC curve with GI cutoff point for prediction of middle-aged (A: >28; Sensitivity = 99.8; specificity = 89.6; area = 0.98; CI (95%) = 0.93–0.99; $p < 0.001$.) and older individuals with low functional autonomy (B: >34; Sensitivity = 99.7; specificity = 87.6; area = 0.97; CI (95%) = 0.95–0.98; $p < 0.001$).

## 4. Discussion

The GDLAM protocol was applied to assess ADL performance in middle-aged and older women in Spain. The tests of this protocol are similar to ADL, such as crossing a street, sitting and getting up from a chair, walking short distances around the house and with changes of direction, dressing, among other activities. This similarity is important since functional autonomy is associated with ADL performance. The components of physical fitness of this protocol involve speed, agility, balance, coordination, power, and resistance of lower and upper limbs in ADL [7,17].

Some previous studies have used the GDLAM protocol to assess functional autonomy levels in elderly patients with chronic obstructive pulmonary disease [18], postmenopausal women [19], postmenopausal women with Parkinson's disease [21], and apparently healthy elderly women [20,22–25], and to analysis the effect of resistance training [19,20,22,25], Pilates [20], aquatic [21,23], and walking exercises [24] interventions. The standard times and the scores provided by the GDLAM protocol to each age group allows the evaluation of Spanish middle-aged and older women. Consequently, the standard values of the present study contribute with accurate standards to categorize functional autonomy and could also be helpful to check the efficiency of exercise training programs [17].

Walking 10 m is important for the individual to have the security and independence to cross a street or move from one room to another in the house. Araújo-Gomes et al. [36] applied resistance exercises (3 days per week) and Pilates (2 days per week) in 12 postmenopausal women (age = 59.83 ± 5.0 years). After 16 weeks of intervention, those women improved the performance in this task (time: 5.71 s). According to the standardized GDLAM protocol classification of the present study, this time is considered "Regular" (5.53–8.00) for the age range of 55 to 59 years and "Good" (5.00–6.65) for the age range of 60 to 64 years. Similarly, in the standardized GDLAM protocol classification of Dantas et al. [15] the age range of 60 to 64 years would also be classified as "Good" (5.52–7.04). However, this classification [15] only included elderly Brazilian women (≥60 years).

Sitting and getting up as well as getting up and moving around the house are movements involved in everyday tasks. For instance, sitting and getting up from a chair or a sofa and going from place to place. Performing these activities satisfactorily requires adequate levels of muscle strength and power,

mainly in the lower limbs, in addition to balance and agility. Likewise, the capacity of standing up from the prone position also requires these physical capabilities. The SCMA, SSP, and SPP tests of the GDLAM protocol are used to analyze these variables [17]. Vale et al. [24] found significant decreases in the time to execute these tasks (SCMA: 44.17 ± 3.08 s; SSP: 9.84 ± 1.56; SPP: 3.01 ± 0.56 s) in a group of 15 elderly women (age: 68 ± 4.4 years) who practiced 24 weeks of resistance exercises (2×/week; 50 min/session).

Putting on and taking off a T-shirt is a daily process throughout life. Sometimes, middle-aged and elderly women present difficulty performing this movement, as they may have some osteoarticular limitation or some functional decrease. Resistance [19,20,22,25] and aquatic exercises [21,23] can improve strength and flexibility. Thus, they can influence the performance of this daily activity, improving functional autonomy. A randomized controlled trial [20] found significant decreases in the time to perform the PTS test in older women from Murcia, Spain, after 36 weeks of interventions (2×/week; 60 min/session). Those positive outcomes were observed both in the Pilates (n = 20; age: 67.5 ± 3.87 years; PTS: 15.53 ± 2.98 s) and the resistance exercises (n = 20; age: 73.36 ± 4.84 years; PTS: 16.24 ± 3.82 s) groups. As reported by the standardized GDLAM protocol classification of the current study, the Pilates and the resistance group are classified as "Regular" (65–69 years: 14.49–20.19 s; 70–74 years: 13.80–20.08 s) in this test.

The five tests of the protocol establish a general indicator of functional autonomy, the GI. The results obtained in the tests of the GDLAM protocol showed strong reproducibility, which increases the precision and accuracy of the data to establish the adequate standardized classification of functional autonomy through the GI. Tests and GI outcomes of the present study showed that, as age increases, the time to perform ADL also increases. This occurs due to the deleterious effects of aging and the lifestyle adopted by individuals, comprising the sedentary behavior [37].

The GI cutoff point provided for prediction of low autonomy can be very useful for healthcare and educational settings. Women with higher scores of the cutoff points should be encouraged to change their lifestyle to prevent dependency in later life. In middle-aged women, GI values above 28 (scores) may indicate the need for specific care and changes in lifestyle to avoid a sedentary lifestyle and aging with low functional autonomy. For older women, GI values above 34 (scores), probably show the necessity for regular physical exercises to maintain ADL. Changes in posture, as one of the aging characteristics, especially from the age of 80, can increase the time of execution of the ADL since, in the present study, an increase in total body mass with lower height was observed in G7. This may be related to the loss of bone mineral density due to aging and intervertebral discs degeneration that causes an increase in spinal curvatures, such as kyphosis intensification [38].

Another important issue in this context is the components of physical fitness that affect health and physical function, such as cardiorespiratory fitness, muscle strength and endurance (muscular fitness), flexibility, neuromotor fitness, and body composition [4]. Therefore, the American College of Sports Medicine (ACSM) [3] and the World Health Organization (WHO) [39] highlight that older people should remain physically active to preserve or even improve skeletal integrity. Ideally, exercise programs for older people should embrace not only weight-bearing endurance and resistance activities to preserve bone mass and to counteract muscle strength decline and muscle mass loss, but also activities intended to increase balance and prevent falls [40–42], which involves ADL performance and functional autonomy.

An active lifestyle can promote healthy aging and greater independence in old age [3,43]. This way, the percentile values adopted to establish the classification of the battery of tests and the GI of the GDLAM protocol can allow the monitoring of individuals to maintain an active lifestyle.

The strengths of the present study are that the reference values of the GDLAM functional autonomy assessment battery did not exist in the Spanish context and the inclusion of a large population sample size, although it is not a representative sample of the Spanish population. As limitations of the study, it must be indicated that it was only conducted with middle-aged and older women. A future line of research is to replicate the study with men and verify if there are differences by sex. Another limitation

is that the number of participants being lower in some of the age groups, due to a lack of interest in participation, which may result in less accurate policy data. This suggests the need to continue researching the motives for practice and motivation of these groups, in order to offer them exercise programs that will achieve their adherence.

## 5. Conclusions

This is the first study providing reference values (or percentiles) for the GDLAM protocol in a large cohort of middle-aged and older Spanish women. Age-specific functional autonomy normative values for middle-aged and older Spanish women have been established. The normative values hereby provided will enable evaluation and adequate interpretation of middle-aged and older Spanish women. This "tool" is especially interesting in healthcare and educational settings for healthy and active aging. Normative values can help clinicians and physical education trainers to know the level of functional autonomy of women practitioners and prescribe exercise programs that fit their level and state of health, to help them improve. In addition, the reported normative values should be used to motivate practitioners to do regular physical activity and increase tFheir functional autonomy for better health and quality of life.

**Author Contributions:** Conceptualization, P.J.M.-P. and R.G.d.S.V.; methodology, P.J.M.-P., N.G.-G., D.J.-P., J.B.P.d.C. and R.G.d.S.V.; software, J.B.P.d.C., R.G.d.S.V.; validation, P.J.M.-P., D.J.-P., J.B.P.d.C. and R.G.d.S.V.; formal analysis, P.J.M.-P., J.B.P.d.C. and R.G.d.S.V.; investigation, P.J.M.-P., N.G.-G., R.V.-C., G.M.G.-G., A.L.-V., A.E.-G., D.J.-P., A.C.-B., D.V.-D., J.B.P.d.C. and R.G.d.S.V.; resources, P.J.M.-P., N.G.-G., R.V.-C., G.M.G.-G., A.L.-V., A.E.-G., D.J.-P., A.C.-B., D.V.-D., J.B.P.d.C. and R.G.d.S.V.; data curation, P.J.M.-P. and R.G.d.S.V.; writing—original draft preparation, P.J.M.-P., N.G.-G., R.V.-C., D.J.-P., J.B.P.d.C. and R.G.d.S.V.; writing—review and editing, P.J.M.-P., N.G.-G., R.V.-C., G.M.G.-G., A.L.-V., A.E.-G., D.J.-P., A.C.-B., D.V.-D., J.B.P.d.C. and R.G.d.S.V.; visualization, P.J.M.-P., N.G.-G., R.V.-C., D.J.-P., D.V.-D., J.B.P.d.C. and R.G.d.S.V.; supervision, P.J.M.-P., N.G.-G., D.J.-P., and R.G.d.S.V.; project administration, P.J.M.-P.; funding acquisition, P.J.M.-P. All authors have read and agreed to the published version of the manuscript.

**Funding:** The present research on active aging of members of HEALTHY-AGE Network (reference 08/UPR/20) is supported by a grant from the Spanish Ministry of Culture and Sport- Sports Sciences Networks and PJMP was supported by a grant from the European Union's ERASMUS + SPORT program (reference: 603121-EPP-1-2018-1-ES-SPO-SCP). DJP was supported by a grant from the Spanish Ministry of Science and Innovation - MINECO (RYC-2014-16938).

**Acknowledgments:** The research team would like to thank the heads of the social and women's centers and all the older women for their participation in this research, and Catholic University San Antonio of Murcia (UCAM) for its support to the line of research on healthy and active ageing.

**Conflicts of Interest:** The authors declare no conflict of interest.

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
