# Peer review of "Functional Autonomy Evaluation Levels in Middle-Aged and Older Spanish Women: On Behalf of the Healthy-Age Network"

_sustainability, doi:10.3390/su12219208_

Round 1
Reviewer 1 Report
The purpose of the manuscript is to determine the functional autonomy levels using the GDLAM protocol in women living in Spain aged between 50 and 90 years old. Moreover, they would develop a classification for this population.
The research design adopted is adequate and the authors conducted the study appropriately. However, I have some comments for the authors.
The background provided by the authors is not sufficient and the different issues presented are not well connected each other. I also suggest focusing the introduction in relation to the purpose of the study (for example, authors stated “The decline in functional autonomy is one of the main losses with aging and 56 can lead to frailty of older individuals” but why deepen this issue?
Please, rewrite this section also considering the recent literature on this topic.
Moreover, although the data analysis performed is suitable, and the results well presented, the authors did not discuss properly and in depth all the findings found. Authors should compare results obtained with other recent outcomes (in agreement and in contrast) reported in the literature.
Finally, authors failed to report strengths and limitations of the study.
SPECIFIC COMMENTS
Line 47-48: “Besides balance, other components of physical fitness that influences health are muscle strength, aerobic capacity, and flexibility”. Authors should be more specific for the population considered in the study. For example, I suggest rewrite the sentence as follows: “Besides balance, other components of physical fitness that influences health are muscle strength, aerobic capacity, and flexibility in adults as well as in older people”
Line 58-60: “A study conducted with 855 community-dwelling older people (53% women) in Spain found a 58 prevalence of frailty of 26%. Moreover, women presented frailty twice as often when compared to 59 men [10].” This sentence does not fit the topic of the manuscript and is not appropriate with the measurements performed. Authors have already explained in the previous sentence that functional autonomy decline can lead to frailty in older individuals. In fact, the three perspectives of autonomy defined in the GLAM and, more generally, the ability of independence do not depend only on fragility. Hence, why argues this issue so thoroughly?
Line 249-252: “Ideally, exercise programs for older people should embrace not only weight-bearing endurance and resistance activities to preserve bone mass, but also activities intended to increase balance and 251 prevent falls [35], which involves ADL performance and functional autonomy”. I suggest to edit this sentence as follow: “Ideally, exercise programs for older people should embrace not only weight-bearing endurance and resistance activities to preserve bone mass and to counteract muscle strength decline and muscle mass loss, but also activities intended to increase balance and 251 prevent falls [35], which involves ADL performance and functional autonomy”. In this sentence I suggest considering and citing the following references:
- Fragala MS, Cadore EL, Dorgo S, Izquierdo M, Kraemer WJ, Peterson MD, Ryan ED. Resistance Training for Older Adults: Position Statement From the National Strength and Conditioning Association. J Strength Cond Res. 2019; 33(8): 2019-2052.
- Battaglia et al.. Walking in Natural Environments as Geriatrician’s Recommendation for Fall Prevention: Preliminary Outcomes from the “Passiata Day” Model. Sustainability. 2020; 12, 2684.
Reviewer 2 Report
This study was conducted to determine the functional autonomy levels using the GDLAM protocol and to develop a classification pattern for middle age and older women living in Spain. As a result of the review:
Introduction section
- Providing relationships between functional autonomy and quality of life may make the introduction more informative.
- You need to provide your hypotheses for the aims of the study.
Materials and Methods section
- You need to explain why you categorized the participants’ age by five years.
- If you collected the data on physical activity levels such as International Physical Activity Questionnaire (IPAQ), it would provide more information in order to understand the importance of physical activities well.
- You need to clarify specific methods to determine cut-off values of GI (e.g. by using Youden’s J statistic).
- This study tried to provide normative values, which are valuable, but it would be better to identify differences in the results of five tests from the GDLAM protocol between age categories in addition to normative values.
Discussion section
- You need to provide more interpretations of the results on each test of the GDLAM protocol because a component of physical fitness that those demonstrate are different.
- You did not write limitations of this study. You need to provide limitations in applying the results of this study to real-world situations. In addition, it would be better to suggest directions of future studies.
[Minor comments]
- You should use full nomenclatures for the ROC curve since you used the term for the first time in this manuscript.
- You should provide the reference of IBM SPSS Statistics 23.
- I recommend that this manuscript should be edited by an English professional editor for more readable. There are several grammatical errors.
Round 2
Reviewer 2 Report
Good!